# DESIGEN: A PIPELINE FOR CONTROLLABLE DESIGN TEMPLATE GENERATION

## ABSTRACT

Templates serve as a good starting point to implement a design (e.g., banner, slide) but it takes great effort from designers to manually create. In this paper, we present **Desigen**, an automatic template creation pipeline which generates background images as well as harmonious layout elements over the background. Different from natural images, a background image should preserve enough non-salient space for the overlaying layout elements. To equip existing advanced diffusion-based models with stronger spatial control, we propose two simple but effective techniques to constrain the saliency distribution and reduce the attention weight in desired regions during the background generation process. Then conditioned on the background, we synthesize the layout with a Transformer-based autoregressive generator. To achieve a more harmonious composition, we propose an iterative inference strategy to adjust the synthesized background and layout in multiple rounds. We construct a design dataset with more than 40k advertisement banners to verify our approach. Extensive experiments demonstrate that the proposed pipeline generates high-quality templates comparable to human designers. More than a single-page design, we further show an application of presentation generation that outputs a set of theme-consistent slides. The data and code will be released.

## 1 INTRODUCTION

Design templates are predefined graphics that serve as good starting points for users to create and edit a design (e.g., banners and slides). For designers, the template creation process typically consists of collecting theme-relevant background images, experimenting with different layout variations, and iteratively refining them to achieve a harmonious composition. Previous works (Zheng et al., 2019; Yamaguchi, 2021; Cao et al., 2022) mainly focus on layout generation conditioned on a given background image and have achieved promising results, but it still remains unexplored to accelerate the handcrafted background creation and the laborious refinement of the background and layout composition. In this paper, we make an initial attempt to establish an automatic pipeline *Desigen* for design template generation. As shown in Figure 1, Desigen can be decomposed into two main components, background generation and layout generation respectively.

Generating an appropriate background is the first step toward successful design creation. Different from natural images, a well-chosen background should contain the context of a design (e.g., theme-related visual content) while leaving necessary non-salient space for overlaying layout elements. It is non-trivial for current state-of-the-art text-to-image (T2I) diffusion models (Ho et al., 2020) to satisfy the above requirements, as they are limited in spatial control for preserving desired space in images with only textual description input. To generate the background, we propose two simple techniques for stronger spatial control, namely salient attention constraint and attention reduction. In our early experiments, we observed that salient objects in the synthesized images are highly correlated with the cross-attention map activation. Therefore, we extend current T2I models with an additional spatial constraint to restrict the attention activation. Specifically, the cross-attention is constrained to approximate the saliency distribution detected from the background during the learning process. Furthermore, we propose a training-free strategy by reducing the attention weights in corresponding regions given the user-specified mask. We will show in our experiments that the enhanced T2I generator can effectively produce high-quality images that can be used as backgrounds.

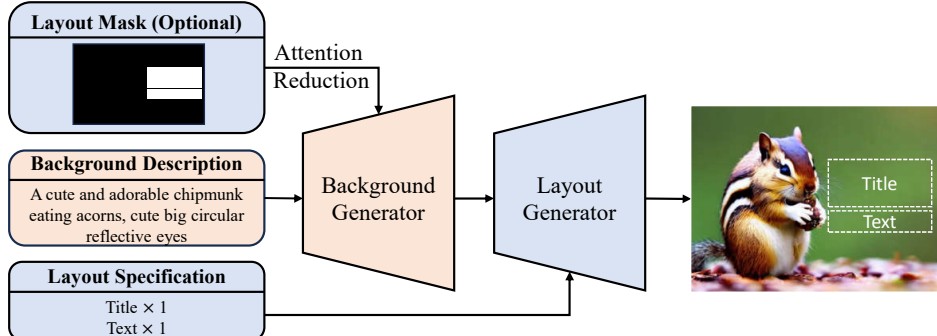

Figure 1: **The process of design template generation.** Desigen first synthesizes the background using a text description. Layout mask is an optional condition to specify regions that should be preserved with non-salient space. Layout is then generated given the background and the specification.

Given the synthesized background, we implement a simple and efficient baseline to generate the subsequent layout. To achieve a harmonious composition between background and layout elements, we propose an iterative inference strategy to further collaborate the background generator with the layout generator. In particular, the salient density in the corresponding layout element region (layout mask in Figure 1) can be lowered by attention reduction to re-generate the background, and the layout can be further refined based on the adjusted background. Such iterations between backgrounds and layouts can be performed multiple times to improve visual accessibility in the final design.

To train and evaluate the proposed pipeline, we construct a dataset of more than 40k banners with rich design information from online shopping websites. For evaluation, we quantitatively measure the background image quality in terms of cleanliness (saliency density), aesthetics (FID), and text-background relevancy (CLIP score). We also evaluate the synthesized layouts with commonly-used metrics such as alignment and overlap, as well as the occlusion rate between layouts and background saliency map to indicate the harmony composition degree. Experiment results show that the proposed method significantly outperforms existing methods. Finally, we show an application of presentation generation to demonstrate that our pipeline can not only generate single-page design, but can also output a series of theme-relevant slide pages.

Our main contributions can be summarized as follows:

- We make an initial attempt to automate the template creation process with the proposed *Desigen* and establish comprehensive evaluation metrics. Moreover, we construct a dataset of advertisement banners with rich design metadata.
- We extend current T2I models with saliency constraints to preserve space in the backgrounds, and introduce attention reduction to control the desired non-salient region.
- To achieve a more harmonious composition of background and layout elements, an iterative refinement strategy is proposed to adjust the synthesized templates.
- We further present an application of Desigen: presentation generation that outputs a set of theme-consistent slides, showing its great potential for design creation.

## 2 RELATED WORK

### 2.1 TEXT-TO-IMAGE GENERATION

Early methods for T2I generation are based on conditional GAN (Reed et al., 2016; Xu et al., 2018; Zhang et al., 2017; 2021; 2018) and auto-regressive architecture (Gafni et al., 2022; Yu et al., 2022). Recently, the rapid rise of diffusion model (Ho et al., 2020; Nichol & Dhariwal, 2021; Dhariwal & Nichol, 2021; Song et al., 2022; Karras et al., 2022) has shown great theoretical potentials to learn a good image distribution. Some diffusion-based T2I models (Ramesh et al., 2022; Saharia et al., 2022; Rombach et al., 2022; Balaji et al., 2022) are successfully boosted to a larger scale and achieve remarkable performance in synthesizing high-quality images. Except for building generative models

with high capability, some works (Choi et al., 2021; Kim et al., 2022; Avrahami et al., 2022) try to edit the synthesized images with frozen pre-trained models, while other methods (Gal et al., 2022; Ruiz et al., 2022) finetune the pre-trained models to generate domain-specific content.

The above methods mainly focus on generic text-to-image generation. Meanwhile, spatial control is also essential for generating desirable images. Most previous works (Gafni et al., 2022; Rombach et al., 2022; Yang et al., 2022b; Fan et al., 2022; Yang et al., 2022a) encode the spatial constraints via an extra condition module, which introduces additional model parameters and requires well-designed training strategy. Diffusion-based editing methods like prompt-to-prompt (Hertz et al., 2022), Diffedit (Couairon et al., 2022), and paint-with-words (Balaji et al., 2022) discover that the cross-attention weights produced by Unet are highly relevant with corresponding text tokens, thus synthesized images can be edited with simple attention mask heuristics. In this paper, we investigate the correlation between the cross-attention weights and the background image saliency map. We propose two techniques of attention map manipulation in both the training and inference phases to generate appropriate backgrounds.

## 2.2 GRAPHIC LAYOUT GENERATION

Previous works on layout generation (Li et al., 2019; Jyothi et al., 2019; Patil et al., 2020; Jiang et al., 2022; Gupta et al., 2021; Kong et al., 2022; Arroyo et al., 2021; Lee et al., 2020; Li et al., 2020; Kikuchi et al., 2021; Weng et al., 2023) mainly focus on a simplified setting which only considers the type and position of layout elements. Most of the methods (Jiang et al., 2022; Gupta et al., 2021; Kong et al., 2022; Arroyo et al., 2021; Kikuchi et al., 2021) flatten the layout elements and generate them by the Transformer architecture as sequence prediction. Although promising results have been achieved by the above methods, content-aware layout generation (e.g., to consider the background) is more useful and closer to real-world scenarios in practice. ContentGAN (Zheng et al., 2019) first incorporates visual content features with a multi-modality feature extractor and utilizes GAN to generate layouts. CanvasVAE (Yamaguchi, 2021) learns the document representation by a structured, multi-modal set of canvas and element attributes. Following works improve the results with heuristic-based modules (Li et al., 2022; Vaddamanu et al., 2022) and auto-regressive decoders (Yamaguchi, 2021; Wang et al., 2022; Cao et al., 2022; Zhou et al., 2022). These works only concentrate on the layout generation stage, while our paper provides an iterative strategy to refine both backgrounds and layouts with a simple but effective layout generator.

## 3 WEB-DESIGN: A DATASET FROM ADVERTISEMENT BANNERS

To the best of our knowledge, there is no current available design dataset with backgrounds, layout metadata, and corresponding text descriptions. To alleviate the data issue, we construct a new design dataset *Web-design* containing 40k online advertising banners with careful filtering.

**Collection.** We crawl home pages from online shopping websites and extract the banner sections, which are mainly served for advertisement purposes with well-designed backgrounds and layouts. The collected banners contain background images, layout element information (i.e., type and position), and corresponding product descriptions. Background images are directly downloaded from the websites while layout elements and content descriptions can be parsed from the webpage attributes.

**Filtering.** While most of the banners are well-designed, we observe that some of them lack visual accessibility where the background salient objects and layout elements are highly overlapped. To improve the dataset quality, we filter the collected banners with the following two criteria: (1) **background salient ratio** (sum of saliency map over total image resolution). We obtain the saliency maps of background images via the ensemble of two popular saliency detection models (Basnet (Qin et al., 2019) and U2Net (Qin et al., 2020)). We only select images with a salient ratio between 0.05 and 0.30 since a large ratio indicates dense objects in the background, while a low ratio below 0.05 is mainly due to detection failure; (2) **occlusion rate** (overlapping degree between background salient objects and layout elements). Banners with an occlusion rate of less than 0.3 are selected, which contributes to a harmonious background and layout composition.

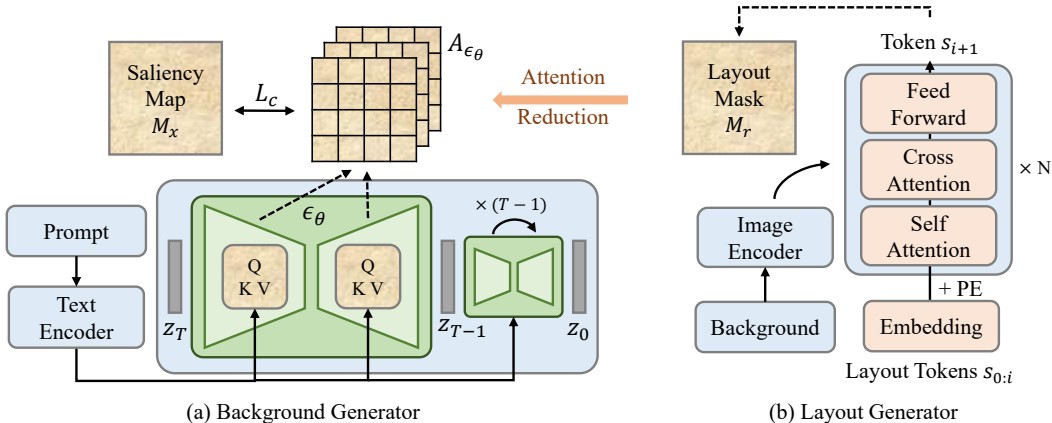

(a) Background Generator                    (b) Layout Generator

Figure 2: **Overview of Design.** (a) background generator synthesizes background images from text descriptions; (b) layout generator creates layouts conditioned on the given backgrounds. By attention reduction, the synthesized backgrounds can be further refined based on input/layout masks for a more harmonious composition.

## 4 METHOD

This section first introduces the background generator with the proposed salient attention constraint and attention reduction control. Then, we present the background-aware layout generator and the iterative inference strategy.

### 4.1 TEXT-TO-IMAGE DIFFUSION MODELS

Recent diffusion-based T2I models (Ramesh et al., 2022; Saharia et al., 2022; Rombach et al., 2022) have achieved remarkable performance in generating images based on arbitrary text prompts. Diffusion model (Ho et al., 2020) is a probabilistic model that learns the data distribution $p(x)$ by iterative denoising from a normal distribution. For every timestep $t$, it predicts the corresponding noise by a denoising autoencoder $\epsilon_\theta(x_t, t); t = 1, 2, ..., T$, where $x_t$ is the denoising variable at timestep $t$. The common objective for diffusion model $L_d$ is to minimize the distance between predicted noise and normal distribution noise added on input $x$:

$$L_d = \mathbb{E}_{x, \epsilon \sim \mathcal{N}(0,1), t} \left[ \|\epsilon - \epsilon_\theta(x_t, t)\|_2^2 \right] \tag{1}$$

**Challenges.** Despite the great capability of large T2I diffusion models, it is difficult to directly apply these models to generate appropriate design backgrounds. They can generate high-quality images but usually are not fit as backgrounds with little space left for placing layout elements. The spatial control is limited with only textual description (e.g., "*leave empty space on the right*"). Moreover, when the background and the layout are generated in a subsequent order, the synthesized image cannot be refined once generated, which prevents further optimization for a more harmonious design composition. Therefore, it still remains unexplored in background generation with spatial control and the refinement of the harmonious background and layout composition.

### 4.2 BACKGROUND GENERATION WITH SPATIAL CONTROL

**Salient Attention Constraint.** Compared with natural images, there should be more non-salient space preserved in the backgrounds. For this purpose, we propose to incorporate the saliency constraint in the attention level. As shown in Figure 3, we observe that dominant pairs (over 90%) of the model attention weights and the detected image saliency map have a high cosine similarity larger than 0.70. This finding indicates that *the salient objects in images are highly correlated with the activation of cross-attention.* Alternatively, space can be well-preserved with the suppression of the cross-attention. Therefore, the attention weights of non-salient regions should be lowered, while the weights of salient regions should be maintained to keep the original context.

We design an auxiliary attention constraint loss $L_c$ to achieve this process:

$$L_c = \mathbb{E}_{x,\epsilon\sim\mathcal{N}(0,1),t}\left[\left\|A_{\epsilon_\theta(x_t,t)} \cdot M_x\right\|\right] \quad (2)$$

where $A_{\epsilon_\theta(x_t,t)}$ is the cross attention map for generating $x_t$, $M_x$ is saliency map of input image $x$ obtained from an external detector in the pre-processing. And the final loss of our background generation model $L_{total}$ would be:

$$L_{total} = L_d + \gamma \cdot L_c \quad (3)$$

where $\gamma$ is the ratio to control the weight of salient constraint loss (set to 0.5 empirically).

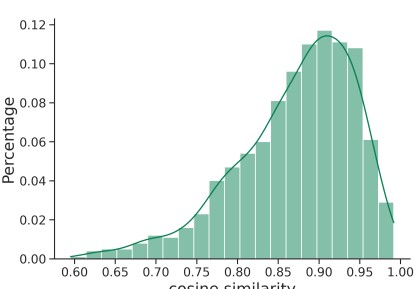

Figure 3: **The cross-attention and saliency maps are highly correlated.**

**Attention Reduction Control.** So far, the background generator can synthesize images with necessary space for overlaying layout elements. However, the region of the preserved space still remains uncontrollable. For stronger spatial controllability, we propose attention reduction, a training-free technique to designate background regions to preserve space. It is achieved by reducing the cross-attention scores of corresponding regions. Specifically, with a user-given region mask $M_r$, the cross-attention weights in $A_{\epsilon_\theta(x_t,t)}$ corresponding to that region are reduced by a small ratio $\beta$:

$$A_{\epsilon_\theta(x_t,t)} = \beta A_{\epsilon_\theta(x_t,t)} \cdot M_r + A_{\epsilon_\theta(x_t,t)} \cdot (1 - M_r) \quad (4)$$

where $t$ is the diffusion time steps ranged from 1 to $T$, $\beta$ can be set to 0.01. This simple technique works well without any additional training cost, further boosting the performance with the previous salient attention constraint.

### 4.3 LAYOUT GENERATION AND ITERATIVE TEMPLATE REFINEMENT

With an appropriate background, the next step is to arrange the position of the elements. Here we implement a simple but effective baseline for background-aware layout generation adapted from LayoutTransformer Gupta et al. (2021). Furthermore, as previous works of layout generation (Zheng et al., 2019; Zhou et al., 2022; Cao et al., 2022) mainly focus on synthesizing layouts conditioned on the background image, they ignore the potential interactive refinement between the background and the layout. To improve the composition, we propose an iterative inference strategy for further refinement.

**Sequential Layout Modeling.** Following Gupta et al. (2021), graphic layouts are formulated as a flatten sequence $s$ consisting of element categories and discretized positions:

$$s = ([bos], v_1, a_1, b_1, h_1, w_1, ..., v_n, a_n, b_n, h_n, w_n, [eos])$$

where $v_i$ is the category label of the $i$-th element in the layout (e.g., text, button). $a_i, b_i, h_i, w_i$ represent the position and size converted to discrete tokens. $[bos], [eos]$ are special tokens for beginning and end. The sequence is modeled by an auto-regressive Transformer (Vaswani et al., 2017) decoder with an image encoder $I(x)$ as shown in Figure 2 (b):

$$p(s) = \prod_{i=1}^{5n+2} p(s_i|s_{1:i-1}, I(x)) \quad (5)$$

The layout generator is trained to predict the next token of the layout sequence at the training stage. During inference, the category tokens are fixed and position tokens are synthesized auto-regressively.

**Iterative Inference Strategy.** Until now, the background image is fixed once generated. To enable a more flexible generation and a more harmonious composition, we propose an iterative refinement during inference using the attention reduction control in section 4.2. Specifically, after a first-round generation of the background and layout, we convert the synthesized layout to a region mask $M_r$ for attention reduction to regenerate the background. This allows the background to be adjusted so that the layout elements are more visually accessible. The refinement strategy can be iterated over multiple times to improve the design quality.

## 5 EXPERIMENT

In this section, we first define our evaluation metrics for background and layout respectively. Then, we show both quantitative and qualitative results of the background generator and layout generator. Finally, we present the slide deck generation as an additional application.

### 5.1 EXPERIMENT SETTING

**Background Evaluation Metrics.** For backgrounds, we design the evaluation metrics in three dimensions: **Salient Ratio** (cleanliness), **FID** (aesthetics), and **CLIP Score** (text-image relevancy).

- **Salient Ratio.** It is used to evaluate if the background preserves enough space for layout elements. We define it as the sum of a saliency map detected from the background image over the total image resolution. The lower the salient ratio, the more space on the images preserved for overlaying layout elements.
- **FID (Heusel et al., 2017).** It measures the distance between the distributions of the background image testset and the generated images. We use the pre-trained Inception model to extract the image feature representations.
- **CLIP Score (Radford et al., 2021).** It measures the similarity of image and textual representations extracted by the pre-trained CLIP model. A higher CLIP score indicates that the generated images are more relevant to the text description.

**Layout Evaluation Metrics.** **Alignment** and **Overlap** are the two most commonly used metrics to evaluate layout quality. Moreover, we define a **Occlusion** score to assess the harmony degree of background and layout composition.

- **Alignment (Li et al., 2020).** Layout elements are usually aligned with each other to create an organized composition. Alignment calculates the average minimum distance in the x- or y-axis between any element pairs in a layout.
- **Overlap (Li et al., 2020) (among layout elements).** It is assumed that elements should not overlap excessively. Overlap computes the average IoU of any two elements in a layout. Layouts with small overlap values are often considered high quality.
- **Occlusion (Cao et al., 2022) (with background saliency map).** Occlusion is defined as the overlapping area between the background saliency map and layout element bounding boxes. It is normalized using the sum of the layout element area. A lower occlusion score indicates that the layout is in less visual conflict with the corresponding background.

**Baselines.** As an initial attempt at text-to-design, there are no baselines that can be directly compared. For background generation, several accessible T2I models are selected as baselines, including **Stable Diffusion** (Rombach et al., 2022) and **DALL-E 2**. (Ramesh et al., 2022). **Stable Diffusion Finetuned** and its fine-tuning (FT) version on the proposed dataset are presented as strong baselines. Note that DALL-E 2 is only used for qualitative comparison and human assessment since the provided web interface is not supported to synthesize a large number of images programmatically. For layout generation, LayoutTransformer (Gupta et al., 2021) and CanvasVAE (Yamaguchi, 2021) are selected as the baselines which are publicly available.

**Implementation Details.** All the training images of the background generator are resized to 512 by 512, while the input image of the layout generator is resized to 224 by 224 due to the image encoder constraints. All images presented in this paper are resized to 3:4 due to the typical background resolutions. We initialize the background generator using Stable Diffusion (Rombach et al., 2022) and train it with loss stated in Equation 3 for 100 epochs with learning rate $1 \times 10^{-5}$ and batch size 32. We train the layout generator for 100 epochs with the learning rate $1 \times 10^{-5}$ and batch size 64. We use early stopping based on validation errors. AdamW (Loshchilov & Hutter, 2017) is used as the optimizer with $\beta_1 = 0.9$ and $\beta_2 = 0.999$.

### 5.2 BACKGROUND GENERATION RESULTS

**Overall Comparison.** We show the overall comparison in Table 1(a). The proposed model achieves the best performance in terms of salient ratio and FID. Specifically, the salient ratio is reduced from

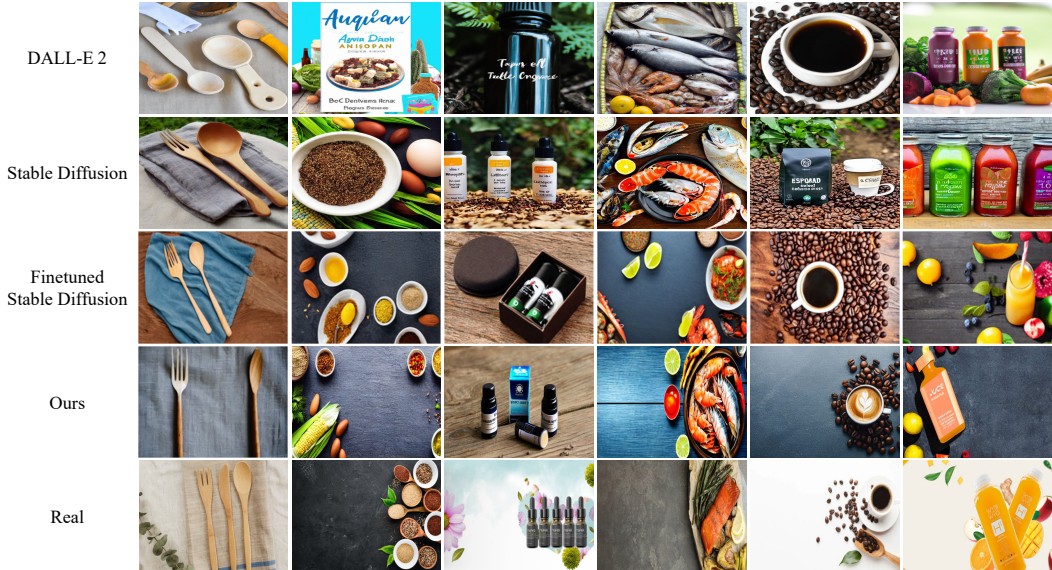

Figure 4: **Generated backgrounds given prompts in the test dataset.** Compared with baselines, our model generates backgrounds with more space preserved, approaching the real designs.

| Method | Salient Ratio | FID | CLIP Score |
|---|---|---|---|
| SD | 35.92 | 39.36 | **31.21** |
| SD (finetuned) | 23.61 | 37.41 | 29.53 |
| Ours ($M_x = \mathbf{1}$) | 21.25 | 34.49 | 29.54 |
| Ours | **20.65** | **31.52** | 29.20 |
| Real Data | 13.86 | - | 27.79 |

| Method | Cleanliness | Aesthetics | Relevancy |
|---|---|---|---|
| DALL-E 2 | 2.98 | 3.34 | 3.43 |
| SD | 3.25 | 3.21 | 3.85 |
| SD (finetuned) | 3.45 | 3.36 | 3.95 |
| Ours | **3.95** | **3.38** | **3.99** |
| Real Data | 5.00 | 5.00 | 5.00 |

(a) Baseline Comparison  (b) Human Assessment

Table 1: **Quantitative results on background image generation.** The metrics are better when they are closer to real data values except for FID (smaller is better).

35.92 to 20.62, with over 50% gain compared with the state-of-the-art T2I model Stable Diffusion (SD), indicating that our synthesized images are more suitable to be backgrounds with enough space for overlaying layout elements. Regarding the CLIP score, our model performance is between SD and real data, which is reasonable since a background needs to balance the trade-off between high text-image relevancy (related objects dominant in the image) and low salient ratio (enough empty space for layout elements). Moreover, we also conduct an ablation setting of the saliency mask in Equation 2 by replacing it using a full mask ($M_x = \mathbf{1}$) to suppress the attention weights in all regions. The full mask provides a strong regularization but increases the difficulty of convergence, leading to worse results compared with the saliency map mask. To further assess the image quality, we conducted a user study inviting 31 ordinary users and professional designers to rate the images from 1 to 5 (larger is better). As shown in Table 1(b), the synthesized images of our model are ranked the highest in all three dimensions, especially in terms of cleanliness.

Next, to validate the effectiveness of the attention reduction control during inference, we take different sizes of masks $M_r$ with random positions to apply in Equation 4. The results are shown in Figure 5. Compared to the model without attention reduction (mask ratio is 0), the performance of the models is significantly improved, which proves the reduction operation is useful. As the mask ratio increases, more spaces are preserved in the images (lower salient ratio). However, the image quality (FID) gets improved before the ratio threshold of 0.5, while the larger ratio hurts the performance since it forces the model to generate an empty background.

**Qualitative Experiments.** Here we show some synthesized background images by current T2I models and ours. As shown in Figure 4, baselines like DALL-E 2 and SD tend to generate images

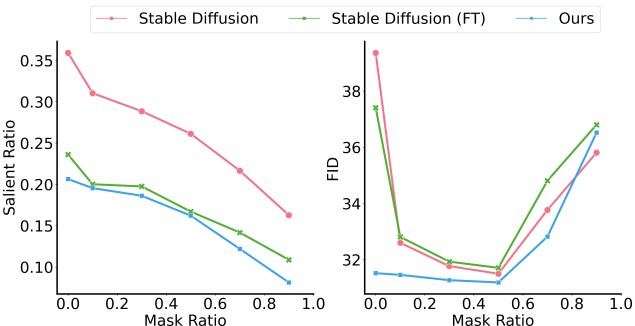

Figure 5: **Synthesized backgrounds with different attention reduction mask sizes.** More space is preserved with higher mask sizes, while the quality is better with the medium mask size.

| Method | Alignment | Overlap | Occlusion |
|---|---|---|---|
| LayoutTransformer | 0.23 | 15.91 | 28.26 |
| CanvasVAE | 1.09 | 20.16 | 13.95 |
| Ours | **0.35** | **14.41** | **13.47** |
| Real Data | 0.33 | 11.32 | 11.89 |

| Method | Alignment | Overlap | Occlusion |
|---|---|---|---|
| Ours-Resnet | **0.31** | 15.43 | 21.44 |
| Ours-ViT | 0.50 | 24.18 | 18.13 |
| Ours-Swin | **0.35** | **14.41** | **13.47** |
| Real Data | 0.33 | 11.32 | 11.89 |

(a) Baseline Comparison

(b) Ablation Study on Image Encoder

Table 2: **Quantitative results on layout generation.** The closer value to the *real data* indicates better generation performance.

with salient objects dominant at the center, which is expected under the pre-trained data distribution. The fine-tuned SD version performs better but is still far from satisfactory. Compared with these methods, the images generated by our proposed model are more appropriate to be backgrounds that have more space preserved for layout elements and are more similar to the real backgrounds.

## 5.3 LAYOUT GENERATION RESULTS

**Overall Comparison.** We compare our layout generator with two different baselines, LayoutTransformer (Gupta et al., 2021) and CanvasVAE (Yamaguchi, 2021). Following the setting in previous works, the performance is better when it is closer to the real data (testset) values. In Table 2(a) and Figure 6, LayoutTransformer performs well at the aspect of alignment and overlap, while failing at producing designs with high visual accessibility. CanvasVAE synthesizes layouts with lower occlusion with backgrounds with relatively low layout quality. Compared with these baselines, the proposed layout generator can generate excellent layouts with good visual accessibility. For the

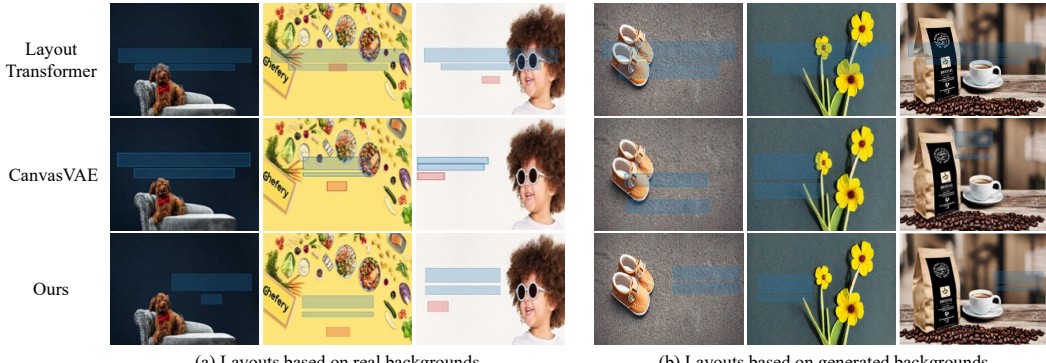

(a) Layouts based on real backgrounds

(b) Layouts based on generated backgrounds

Figure 6: **Comparison with different baselines in layout generation.** The blue boxes represent text elements and the red boxes represent button elements.

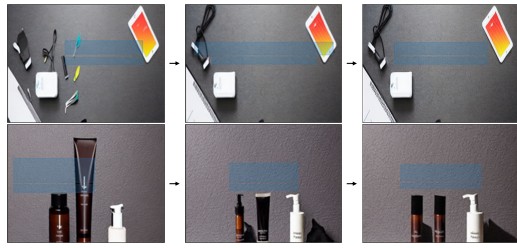

Figure 7: **Qualitative results for iterative design refinement.**

| Iteration | Salient Ratio | Occlusion |
|:---:|:---:|:---:|
| 1 | 16.87 | 11.66 |
| 2 | 15.66 | 8.96 |
| 3 | **15.27** | **8.69** |

Table 3: **Quantitative results for iterative design refinement.**

ablation study, we try different image encoder architectures for background conditioning, including Resnet (Resnet-50) (He et al., 2016), ViT image encoder from CLIP (ViT-base) (Radford et al., 2021) and Swin Transformer (Swin-base) (Liu et al., 2021). As shown in Table 2(b), with Swin Transformer as the visual backbone, the layout generator achieves the best, especially regarding the occlusion metric.

**Iterative Design Refinement.** As mentioned in section 4.3, our pipeline supports iterative refinement between background and layout to achieve a more precise and harmonious composition. Figure 7 shows a refinement process and Table 3 shows the corresponding metrics. After obtaining a background and a layout (1st column), our pipeline inverts the background image to the initial latent, produces a region mask with the generated layout as attention reduction input, and re-generates the background image. Conditioned on the refined background, layouts are also adjusted for better visual accessibility. As shown in the figure, the final design composition looks more harmonious after two iterations of refinement.

## 5.4 PRESENTATION GENERATION

Slide templates within a deck should consistently follow the same theme, with theme-relevant backgrounds and a set of diverse layouts (e.g., title page, title + content page). We demonstrate the ability of our pipeline to create slide decks in Figure 8 (each deck per row). To achieve this, we take the same prompt along with different pre-defined layouts as mask input with the same random seed. Using attention reduction control, different space regions on backgrounds are preserved corresponding to the layout masks, while all background images still contain theme-relevant contexts with deck consistency.

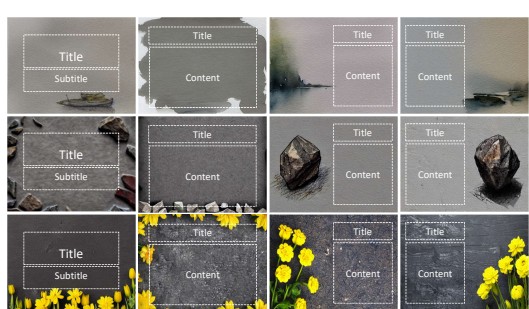

Figure 8: **Presentation generation.**

## 6 CONCLUSION

In this paper, we make an initial attempt to automate the design template creation process. We propose a pipeline **Desigen**, consisting of a background generator and a layout generator. Specifically for background generation, we extend current large-scale T2I models with two simple but effective techniques via attention manipulations for better spatial control. Coupled with a simplified layout generator, experiments show that our pipeline generates backgrounds that flexibly preserve non-salient space for overlaying layout elements, leading to a more harmonious and aesthetically pleasing design template. In the future, we plan to equip our pipeline with more guidance from graphic design principles. Also, we will consider more diverse types of visual assets for enriching design templates, such as decorations and fonts.

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

# A APPENDIX

## A.1 DESIGN DATASET

To the best of our knowledge, there is no current available dataset including content descriptions, background images, and layout metadata. Therefore, we crawl home pages from online shopping websites and extract the banner sections, which are mainly served for advertisement purposes with well-designed backgrounds and layouts, leading to a new dataset of advertising banners. The collected banners contain background images, layout element information (i.e., type and position), and text descriptions. Several sample designs are shown in Figure 9. In Figure 10, we demonstrate the word cloud of the top 200 words of corresponding text descriptions, showing that descriptions are highly relevant to the graphic designs.

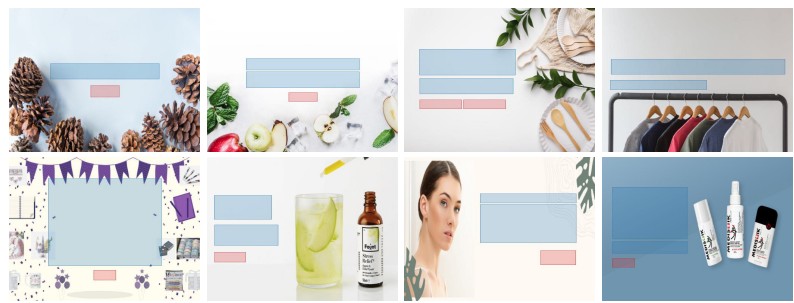

Figure 9: Samples of design from the proposed dataset.

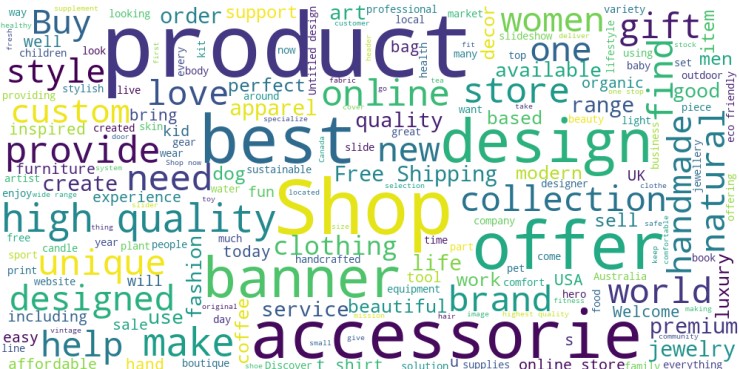

Figure 10: Word cloud of the top 200 words of text descriptions on the proposed design dataset.

## A.2 MORE QUALITATIVE RESULTS

In this section, we show the additional results for the proposed model. Figure 12 shows the additional generated banners via the proposed layout generator, where the blue boxes represent text elements, the red boxes represent button elements and the green boxes represent image elements. It shows that our layout generator performs well on both real backgrounds and generated backgrounds. In Figure 11, the proposed model can synthesize varying background images given the same text prompts. In Figure 15, additional results are shown for slide deck generation with the same prompts and different attention reduction masks. Figure 14 shows additional attention visualization of background-aware layout generation. The layout generator highly attends to the salient objects of the background images and creates layouts on the region with low attention scores.

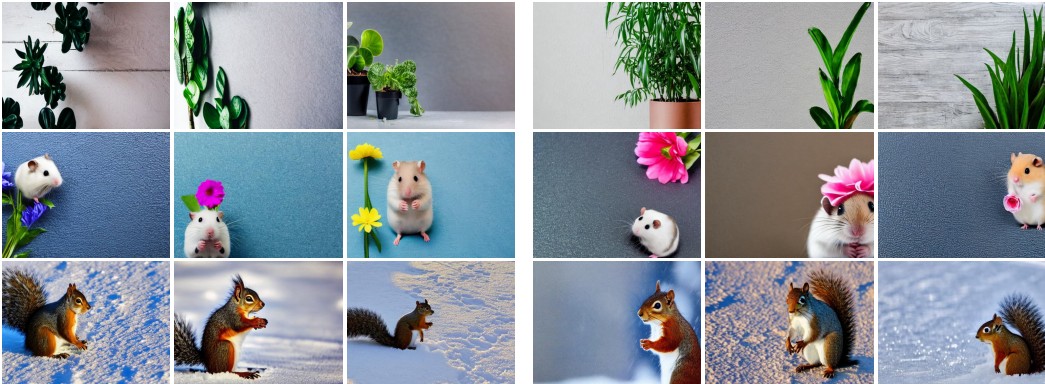

Figure 11: The proposed model can synthesize varying background images given the same text prompts. The prompt of the first row: green plants with white background; the second row: Cute little hamster with flower; the third row: Squirrel in the snow.

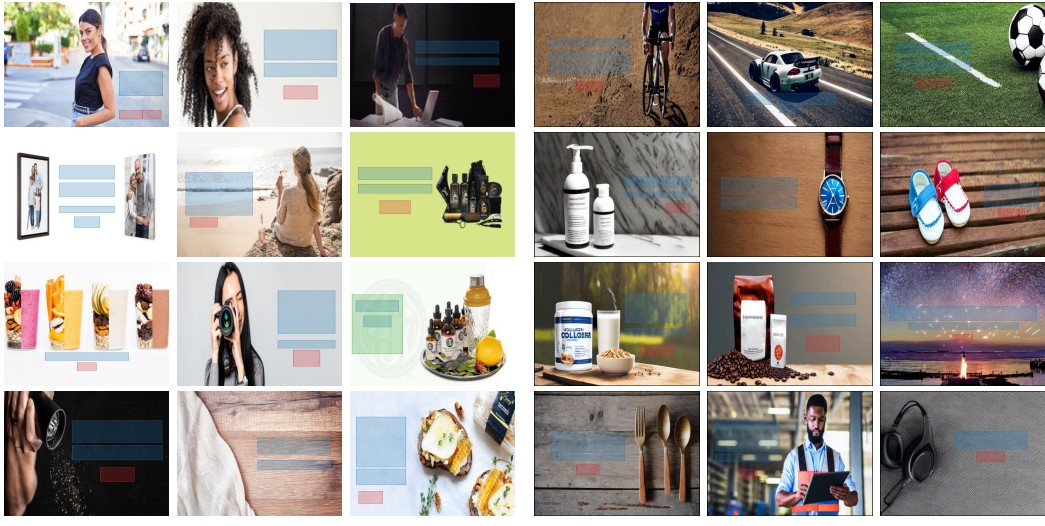

(a) Designs based on real background          (b) Designs based on generated background

Figure 12: More synthesized layouts via the proposed layout generator, where the blue boxes represent text elements, red boxes represent button elements and green boxes represent image elements.

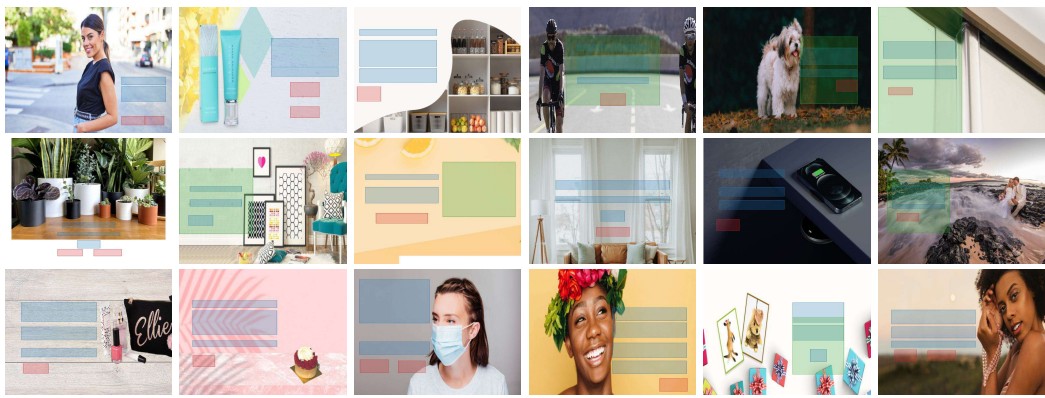

Figure 13: More qualitative results of complicated layouts (more than four elements), where the blue boxes represent text elements, red boxes represent button elements, and green boxes represent image elements.

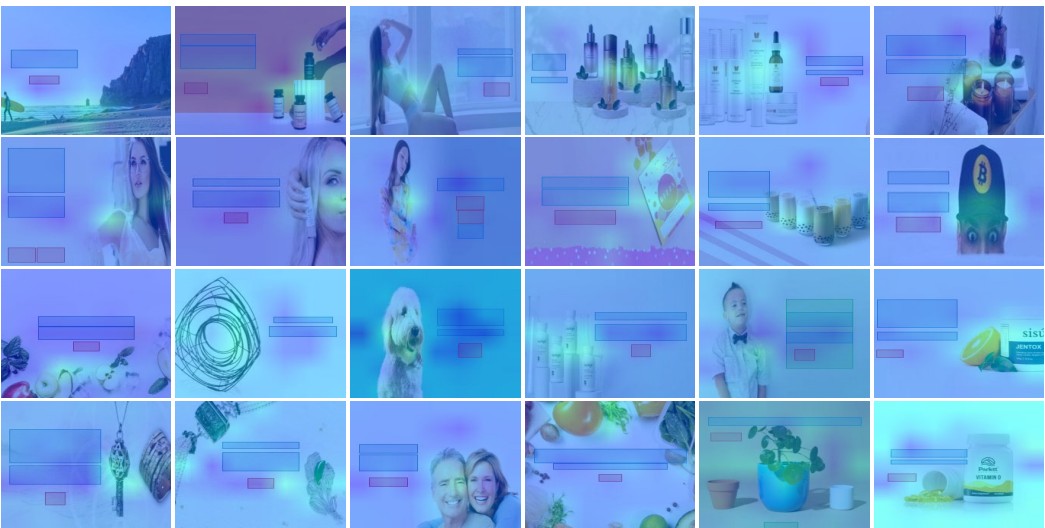

Figure 14: Additional attention visualization of background-aware layout generation. The layout generator highly attends to the salient objects of the background images and creates layouts on the region with low attention scores.

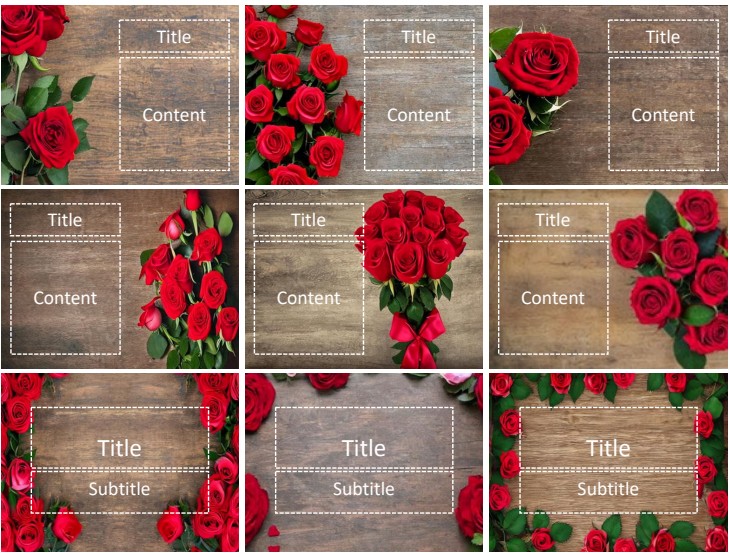

Figure 15: Additional results for slide deck generation with the same prompts and different attention reduction masks (Prompt: Rose for Valentine's Day).

