# OpenReview forum: "Desigen: A Pipeline for Controllable Design Template Generation"
_ICLR.cc/2024/Conference — ICLR 2024 Conference Withdrawn Submission_

### Official Review · Reviewer_umPG · 2023-10-30

**Soundness:** 1 poor
**Presentation:** 2 fair
**Contribution:** 2 fair
**Rating:** 3
**Confidence:** 4

**Summary:**

The focus of the paper is on automating the process of design template generation.

The main contribution is a design template generation pipeline, consisting of two stages: background generation and layout generation. First, a background image is generated with an extended T2I diffusion model that imposes saliency constraints on the cross-attention activations to preserve space for subsequent layout elements. Then, a layout on the generated background is generated with an autoregressive Transformer, which is then refined together with the background in an alternating fashion for a harmonious composition.

A large-scale banner dataset with rich annotations is constructed for training and testing the method, and an application of the method to multi-page design template generation is demonstrated.

**Strengths:**

1. Automatic generation of design templates is an important problem to study.

2. The constructed dataset, if can be publicly released, will be of value to the graphic design synthesis community.

**Weaknesses:**

1. The scale of technical novelty is limited. First, the relationship between subjects in the generated image and their attention activations has already been explored in a recent work, Attend-and-Excite. In view of this work, the finding at the beginning of Sec. 4.2 (and in  Fig.3) is unsurprising and the high-level idea of modifying attention values to control the generated images is not new. Second, the proposed spatial control strategy in Sec. 4.2 (including salient attention constraint and attention reduction) is simple and straightforward, which does not bring much novel technical insight. Third, the layout generator is just a previously proposed technique, i.e., LayoutTransformer. More importantly, the proposed iterative inference strategy for background and layout refinement seems to be ad-hoc. It would be desirable to propose a more unified, perhaps learning-based, approach to capture the dependency between the background and layout, e.g., by modeling their joint distribution, which will be of more interest to the ICLR community.

2. The evaluation is insufficient. For background generation, the diversity of generated images is not evaluated. For layout generation, more evaluation metrics such as FID and max IoU as in [Kikuchi et al., 2021] should be used but are missing. Furthermore, background and layout are now evaluated separately, but the overall design template, which is formed by composing the two, is not evaluated. This is unreasonable since the paper is claimed to be aimed for design template generation instead of background or layout generation. Thus, an evaluation on the quality of generated design templates, e.g., at least through some human studies, is needed to support the paper’s main claim, which is missing in the current paper.

**Questions:**

How is the attention map A obtained in Eq. (2)?

---

> ### Author Response · Authors · 2023-11-12
> **Reply to Reviewer umPG**
>
> ## Technical Novelty
>
> > The novelty of background generation is limited since the relationship between attention activation and subjects is explored in recent work Attend-and-Excite.
>
> Though the similar analysis process, there are several key differences between our background generator and the recent work Attend-and-Excite:
>
> - The motivation is different. Attend-and-Excite is a general method to solve catastrophic neglect (fails to generate one or more of the subjects from the input prompt) while our background generator aims at preserving space to overlay layouts for controllable design generation.
>
> - The proposed method is different. Attend-and-Excite applies a Gaussian kernel on each attention map to obtain smoothed attention maps. Our method introduces the saliency map as an additional training constraint for background and reduces the activation of the corresponding region at inference.
>
> - The applied stage is different. We propose salient attention constraint for training and attention reduction for inference while Attend-and-Excite is an inference-only method.
>
>
> > The layout generator is just a previously proposed technique, i.e., LayoutTransformer.
>
> The key difference between our work and LayoutTransformer is that we enable the layout generator to be background-aware via a simple yet effective cross-attention mechanism. In the paper, Table 2 and Figure 6 show that the proposed layout generator outperforms LayoutTransformer by a large margin, as well as the VAE-based method CanvasVAE. This shows the proposed layout generator can serve as an effective baseline for future content-aware layout generation studies.
>
> > The proposed iterative inference strategy for background and layout refinement seems to be ad-hoc.
>
> - It is natural for human designers to iteratively adjust the background and layout for better visual presentation. Compared with the single feed-forward generation, our iterative strategy allows the refinement between background and layout, leading to a more harmonious composition as shown in Table 3 and Figure 7.
>
> - The idea of modeling the joint distribution of background and layout sounds interesting but much more challenging. For instance, a larger scale of design dataset and more computation resources are required for such joint modeling.
>
>
> > How is the attention map A obtained in Eq. (2)?
>
> In the forward process of denoising U-Net, the cross-attention activations are calculated and stored for the salient loss computation.
>
> ## Design Evaluation
>
> > The evaluation of the design is insufficient.
>
> The numerical evaluation of the generated designs is an important but challenging task. In this paper, we try to build a comprehensive evaluation system for synthesized designs. We use 6 metrics to evaluate the design, including Salient Ratio (cleanliness), FID (aesthetics), and CLIP Score (text-image relevancy) for background, as well as Alignment, Overlap, and Occlusion to evaluate the layout and its combination with background.
>
> > Background diversity is not evaluated.
>
> - Figure 11 of supplementary shows the background diversity given the same text prompt.
>
> - Figure 5 of supplementary shows the background diversity given the same text prompt and different attention reduction masks.
>
>
> > Evaluation metrics of layouts are missing.
>
> In Table 2, we have provided the typical metrics like overlapping and alignment, which are already enough to numerically show the quality of the generated layout.
>
> > The evaluation of the overall design is missing.
>
> We use the Occlusion metric to evaluate the combination of background and layout, which can be regarded as an overall metric for design. This metric is also well aligned with human evaluation.

---

### Official Review · Reviewer_14nH · 2023-11-01

**Soundness:** 4 excellent
**Presentation:** 4 excellent
**Contribution:** 2 fair
**Rating:** 6
**Confidence:** 3

**Summary:**

The paper generates single-image design templates, e.g., for slides and advertisements, based on a generator trained to do so, given an image prompt layout for which parts of the image should be empty.  The paper notes that the cross-attention values often correspond to saliency, providing a direct way to control saliency.

**Strengths:**

The paper solves a novel and useful task. The results seem promising. The cross-attention/saliency observation is interesting.

**Weaknesses:**

I'm not sure that this task is necessarily of great interest to the ICLR community; it may be better-suited for another venue.

The layout-generation component of the work does not seem too novel and is not compared to the state-of-the-art:

PosterLayout: A New Benchmark and Approach for Content-Aware Visual-Textual Presentation Layout
Hsiao Yuan Hsu, Xiangteng He, Yuxin Peng, Hao Kong, Qing Zhang; Proceedings of the IEEE/CVF Conference on Computer Vision and Pattern Recognition (CVPR), 2023, pp. 6018-6026


For the image-generation component, much of the same effect can be achieved with existing diffusion models. For example, I tried prompts like `“a background image with empty space for a shoe advertisement”` or `"a background image of a squirrel with empty space for advertising text"`, I got reasonable results with empty spaces. This doesn't offer the same level of control as the proposed system, but seems much simpler, and one could run many generations to find reasonable layouts.

**Questions:**

None

---

> ### Author Response · Authors · 2023-11-12
> **Reply to Reviewer 14nH**
>
> > For the image-generation component, much of the same effect can be achieved with existing diffusion models by enhancing prompts like like `a background image with empty space`.
>
> Although similar effect can be achieved with existing models, our background generator is more controllable and perform consistently well. To further demonstrate the effectiveness of our model, we show the baseline performance by enhancing context prompts with background-specific keywords (e.g., background image, with empty space to overlay layouts) with Stable Diffusion.
>
> | Method | Salient Ratio | FID | CLIP Score |
> | :---: | :---: | :---: | :---: |
> | SD (context prompt) | 35.92 | 39.36 | 31.21 |
> | SD (enhanced prompt) | 26.54 | 41.54 | 30.49 |
> | Ours (context prompt) | 20.65 | 31.52 | 29.20 |
>
> This table shows that, with enhanced prompts, the salient ratio is lower (which means there is more space preserved). However, the FID score becomes worse, which means the image quality may be damaged using such background-specific keywords. Instead, with the proposed method with only context prompts, both the salient ratio and FID become better, showing its effectiveness on background generation.
>
> > The layout-generation component of the work does not seem too novel.
>
> - Designing a novel layout generator is not the main purpose of this paper. Instead, our key contribution to layout generation is to provide an iterative refinement strategy. Such an architecture-free strategy can further improve the visual harmony of the background and layout.
>
> - The performance of the proposed layout generator is shown in Table 2, which indicates that such architecture is simple yet effective and works surprisingly well.

---

### Official Review · Reviewer_XhSd · 2023-11-02

**Soundness:** 2 fair
**Presentation:** 2 fair
**Contribution:** 2 fair
**Rating:** 5
**Confidence:** 3

**Summary:**

This paper aims to devise a model that could automatically generate a background image with the layout of multiple elements on the image, which could be useful for designing templates for slides, ads, webpages, etc. The desired properties of such templates are: i) background image should leave enough space for overlaying elements (like texts); ii) texts and titles should fit proportionally in the blank space on the image, not occluding each other. To achieve the above goal, the proposed method first generates an initial background with with spatial control using cross-attention on saliency maps, and then a transformer based layout generation model that iteratively refines the background image and the positions and proportions of the elements on it. Qualitative and quantitative results were presented to show that the proposed method generates images that are more suitable as background and also generates layouts with more harmonious elements than baseline methods.

**Strengths:**

- This work proposes clear and extensive automatic metrics to evaluate the success of layout generation (salient ration, alignment, overlap, occlusion).
- After defining the goal as generating a background image with enough blank space for overlaying texts, the proposed method is effective in achieving the goal.
- Ablation study for each proposed component is reported.

**Weaknesses:**

- If more discussions can be included on exactly what prompts were experimented for baselines like DALL-E2 and Stable Diffusion, it would really help me understand what existing work is missing that this work provides. Currently the paper reads like baselines and proposed method use the same prompts to generate background images. Is that a fair setting? Baseline models were not trained to generate background only images, so without giving them more explicit prompts specifying that the output should should be a background, it seems natural to me that the output images from baselines are not suitable for background.
- I wonder why the number of elements in a layout (and their minimal and maximal allowed size) is not an input to the background image generation model? It seems to me that the space to leave blank and its position depends on the potential space the elements would take, so this can be a crucial context for the generation model. While an iterative refinement mechanism is used to adjust the background image and the layout, the iterative nature can be confining the model to only make improvements in a local range of the initial image.

**Questions:**

- What was the prompts accompanying each image in the newly collected Web-design dataset, and how were they obtained?
- In Section 5.1 Implementation Details, it mentions that all training images are resized to 512x512. Does this mean that during inference time the model also generates images of size 512x512? It seems to me that advertisement images can come in a wide range of aspect ratios, would the resizing and squared output size limit the use case of this model?
- From the qualitative examples, it seems like each layout can have different numbers of elements on it. How are the elements in each layout determined?

---

> ### Author Response · Authors · 2023-11-12
> **Reply to Reviewer XhSd**
>
> > How is the prompt constructed in background generation experiments? Is it better to enhance the prompt with background-specific keywords for fair comparison?
>
> For evaluation, we use the prompts from the test split of the proposed dataset. Most of these prompts only contain the context in the background (i.e., context prompt). To further demonstrate the effectiveness of the proposed background generator, we show the baseline performance by enhancing context prompts with background-specific keywords (e.g., background image, with empty space to overlay layouts) with Stable Diffusion.
>
> | Method | Salient Ratio | FID | CLIP Score |
> | :---: | :---: | :---: | :---: |
> | SD (context prompt) | 35.92 | 39.36 | 31.21 |
> | SD (enhanced prompt) | 26.54 | 41.54 | 30.49 |
> | Ours (context prompt) | 20.65 | 31.52 | 29.20 |
>
> This table shows that, with enhanced prompts, the salient ratio is lower (which means there is more space preserved). However, the FID score becomes worse, which means the image quality may be damaged using such background-specific keywords. Instead, with the proposed method with only context prompts, both the salient ratio and FID become better, showing its effectiveness on background generation.
>
> > Why the number of elements in a layout (and their minimal and maximal allowed size) is not an input to the background image generation model?
>
> - First, it increases the ambiguity and difficulty of background modeling. The empty area in the background is not directly correlated with the number of elements, which makes the modeling more ambiguous. Furthermore, it is difficult to inject such low-frequent conditions (e.g., the number of elements) into the pre-trained image diffusion models.
>
> - Second, it is not quite necessary. For the application of iterative refinement, if local editing is desired, a solution is to pass in an initial attention reduction mask for the background generator. Then the next-round refinement would be expected to be minor in this way.
>
>
> > How were the prompts collected in the Web-design dataset?
>
> The prompts come from the alt text of background images and the `description` tag of the shopping webpage. We also parse some meaningful words from the background filename as a supplement. For more details of the prompts, Figure 10 in supplementary shows the word cloud of the top 200 words of the prompts on the proposed design dataset.
>
> > How to generate different ratios of backgrounds at inference?
>
> Actually, Stable Diffusion can support various resolutions (although trained on fixed 512x512). We resize all the images (mostly in 3:4) in the dataset into 512x512 resolution for more efficient training. In inference, we generate 512x512 backgrounds and resize them back to 3:4 resolution. If other ratios of backgrounds are desired, users can just generate different resolutions of backgrounds and resize them correspondingly. Inference-only techniques like (Multidiffusion, ICML'23) can also be used to improve the image quality for such scenario.
>
> > How are the elements in each layout determined?
>
> There are two ways to determine the number of elements:
>
> - Decided by layout generator: Since the layout tokens are generated via an autoregressive Transformer, they reach their end when a <BOS> token is generated.
>
> - Decided by user input: Each layout element is formulated by 1 category token plus 4 position tokens. The number of layout elements can be controlled by providing the category tokens and only the position tokens are generated for each layout element.
>
>
> In our experiments, we use the latter way to control the number of layout elements (the number and type of category tokens come from the test dataset).